# Unsupervised Protein-Ligand Binding Energy Prediction via Neural Euler's Rotation Equation

**Wengong Jin, Siranush Sarzikova, Xun Chen, Nir Hacohen, Caroline Uhler**

Broad Institute of MIT and Harvard

`{wjin,sarkizov,xun,nhacohen,cuhler}@broadinstitute.org`

## Abstract

Protein-ligand binding prediction is a fundamental problem in AI-driven drug discovery. Previous work focused on supervised learning methods for small molecules where binding affinity data is abundant, but it is hard to apply the same strategy to other ligand classes like antibodies where labelled data is limited. In this paper, we explore unsupervised approaches and reformulate binding energy prediction as a generative modeling task. Specifically, we train an energy-based model on a set of unlabelled protein-ligand complexes using SE(3) denoising score matching (DSM) and interpret its log-likelihood as binding affinity. Our key contribution is a new equivariant rotation prediction network for SE(3) DSM called Neural Euler's Rotation Equations (NERE). It predicts a rotation by modeling the force and torque between protein and ligand atoms, where the force is defined as the gradient of an energy function with respect to atom coordinates. Using two protein-ligand and antibody-antigen binding affinity prediction benchmarks, we show that NERE outperforms all unsupervised baselines (physics-based potentials and protein language models) in both cases and surpasses supervised baselines in the antibody case.

## 1 Introduction

One of the challenges in drug discovery is to design ligands (small molecules or antibodies) with high binding affinity to a target protein. In recent years, many protein-ligand binding prediction models have been developed for small molecules [13, 19, 43]. These models are typically trained on a large set of crystal structures labeled with experimental binding data. However, it is difficult to apply this supervised learning approach to other ligand classes like antibodies where binding affinity data is limited. For example, the largest binding affinity dataset for antibodies [31] has only 566 data points. Transfer learning from small molecules to antibodies is also hard as their structures are very different.

Given this data scarcity challenge, we aim to develop a general unsupervised binding energy prediction framework for small molecules and antibodies. The basic idea is to learn an energy-based model (EBM) [16] of protein-ligand complexes by maximizing the log-likelihood of crystal structures in the Protein Data Bank (PDB). This generative modeling approach is motivated by recent advances of protein language models (PLMs) which show that the likelihood of protein sequences is correlated with protein mutation effects [20, 11]. However, PLMs are not applicable to small molecules and they model the likelihood of protein sequences rather than structures. We suppose the likelihood of protein complex structures is a better predictor of binding affinity, which is determined by the relative orientation between a protein and a ligand. Indeed, our method shares the same spirit as traditional physics-based energy functions [1] designed by experts so that crystal structures have low binding energy (i.e., high likelihood). Our key departure is to use neural networks to learn a more expressive energy function in a data-driven manner. Importantly, our model is applicable to large-scale virtual screening because it does not require expensive molecular dynamic simulations [21].

37th Conference on Neural Information Processing Systems (NeurIPS 2023).

Specifically, we train EBMs using SE(3) denoising score matching (DSM) [33, 15] as maximum likelihood estimation is hard. For each training example, we create a perturbed protein-ligand complex by randomly rotating/translating a ligand and ask our model to reconstruct the rotation/translation noise. The main challenge of SE(3) DSM in our context is how to predict rotation noise from the score of an EBM in an equivariant manner. Motivated by physics, we develop a new equivariant rotation prediction network called Neural Euler's Rotation Equation (NERE). The key observation is that the score of an EBM equals the force of each atom, which further defines the torque between a protein and a ligand. Using Euler's rotation equation, we convert this torque into the angular velocity of a ligand and its corresponding rotation matrix to calculate SE(3) DSM loss. Importantly, NERE is compatible with any SE(3)-invariant networks and guarantees equivariance without any further requirement on network architectures.

We evaluate NERE on two protein-ligand and antibody-antigen binding affinity benchmarks from PDBBind [36] and Structural Antibody Database (SAbDab) [31]. We compare NERE with unsupervised physics-based models like MM/GBSA [21], protein language models (ESM-1v [20] and ESM-IF [11]), and a variety of supervised models regressed on experimental binding affinity data. To simulate real-world virtual screening scenarios, we consider two settings where input complexes are crystallized or predicted by a docking software. NERE outperforms all unsupervised baselines across all settings and surpasses supervised models in the antibody setting, which highlights the benefit of unsupervised learning when binding affinity data is limited.

## 2   Related Work

**Protein-ligand binding models** fall into two categories: supervised and unsupervised learning. Supervised models are trained on binding affinity data from PDBBind [36, 29, 3, 32, 40, 19, 43]. They typically represent protein-ligand 3D complex represented as a geometric graph and encode it into a vector representation using a neural network for affinity prediction. Unsupervised models are either based on physics or statistical potentials. For example, molecular mechanics generalized Born surface area (MM/GBSA) [21] calculates binding energy based on expensive molecular dynamics. Our work is closely related to statistical potentials like DrugScore$_{2018}$ [6]. It learns an energy score for each atom pair independently based on their atom types and distance alone, which ignores the overall molecular context. Our key departure is to use neural networks to learn a more expressive energy function powered by context-aware atom representations.

**Antibody-antigen binding models** are traditionally based on scoring functions in protein docking programs [25, 1, 10, 39, 2]. They are typically weighted combinations of physical interaction terms whose weights are tuned so that crystal structures have lower energy than docked poses. Recently, protein language models like ESM-1v [20] and ESM-IF [11] have been successful in predicting mutation effects for general proteins and we evaluate their performance for antibodies in this paper. Due to scarcity of binding affinity data, very few work has explored supervised learning for antibody binding affinity prediction. Existing models [23] are based on simple hand-crafted features using binding affinity data from SAbDab. For fair comparison, we implement our own supervised baselines using more advanced neural architectures [38, 28].

**Denoising score matching** (DSM) is a powerful technique for training score-based generative models [33] and has been applied to protein structure prediction [41] and protein-ligand docking [5]. Our departure from standard DSM is two fold. First, standard DSM is based on Gaussian noise while our method uses rigid transformation noise based on an isotropic Gaussian distribution for SO(3) rotation group [15]. Second, we parameterize the score as the gradient of an energy-based model rather than the direct output of a neural network. We choose this parameterization scheme because we need to predict energy and score at the same time.

**Equivariant rotation prediction**. Motivated by molecular docking, there has been a growing interest in predicting rotations with equivariant neural networks. For instance, Ganea et al. [9], Stärk et al. [35] developed a graph matching network that first predicts a set of key points between a protein and a ligand, and then uses the Kabsch algorithm [14] to construct a rigid transformation. Corso et al. [5] predicted rotations based on tensor field networks [37] and spherical harmonics. Our method (NERE) has two key differences with prior work. First, it does not require key point alignment and directly outputs an rotation matrix. Second, NERE does not require spherical harmonics and can be plugged into any SE(3)-invariant encoder architectures for equivariant rotation prediction.

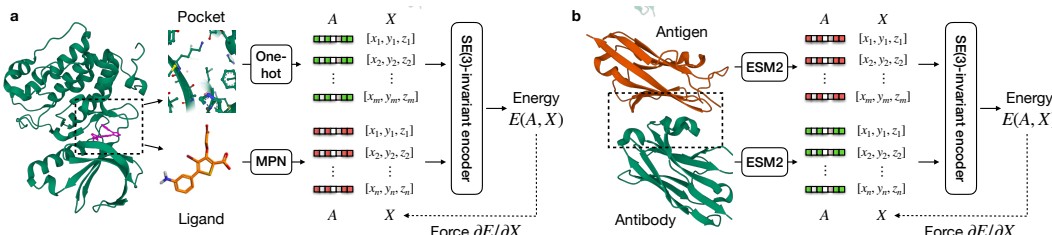

Figure 1: The architecture of $E(\boldsymbol{A}, \boldsymbol{X})$ for small molecules (left) and antibodies (right). For small molecules, the input is a binding pocket and a molecular graph encoded by a message passing network (MPN). For antibodies, the input contains antibody CDR residues and epitope residues, where residue features $\boldsymbol{A}$ come from an ESM2 protein language model.

## 3 Unsupervised Binding Energy Prediction

A protein-ligand (or an antibody-antigen) complex is a geometric object that involves a protein and a ligand (a small molecule or an antibody). It is denoted as a tuple $(\boldsymbol{A}, \boldsymbol{X})$, with atom features $\boldsymbol{A} = [\boldsymbol{a}_1, \cdots, \boldsymbol{a}_n]$ and atom coordinates $\boldsymbol{X} = [\boldsymbol{x}_1, \cdots, \boldsymbol{x}_n]$ (column-wise concatenation). The key property of a complex is its binding energy $E(\boldsymbol{A}, \boldsymbol{X})$. A lower binding energy means a ligand binds more strongly to a protein. In this section, we describe how to parameterize $E(\boldsymbol{A}, \boldsymbol{X})$ and design proper training objectives to infer the true binding energy function from a list of crystal structures without binding affinity labels.

### 3.1 EBM Architecture

An energy function $E(\boldsymbol{A}, \boldsymbol{X})$ is composed of a protein encoder and an output layer. The encoder is a frame averaging neural network [28] that learns a SE(3)-invariant representation $\boldsymbol{h}_i$ for each atom. We choose this architecture because of its simplicity (detailed in the appendix). The output layer $\phi_o$ is a feed-forward neural network with one hidden layer. It predicts the interaction energy $\phi_o(\boldsymbol{h}_i, \boldsymbol{h}_j)$ for each pair of atoms. Finally, we define $E(\boldsymbol{A}, \boldsymbol{X})$ as the sum of pairwise interaction energies:

$$E(\boldsymbol{A}, \boldsymbol{X}) = \sum_{i,j:d_{ij}<d} \phi_o(\boldsymbol{h}_i, \boldsymbol{h}_j) \tag{1}$$

Since atomic interaction vanishes beyond certain distance, we only consider atom pairs with distance $d_{ij} < d$. So far, we have described our method in generic terms. We now specify the input features and preprocessing steps tailored to small molecules and antibodies.

**Small molecules**. When the ligand is a small molecule, the input to our model is a protein-ligand complex where the protein is cropped to its binding pocket (residues within 10Å from the ligand). The model architecture is illustrated in Figure 1a. On the protein side, each atom in the binding pocket is represented by a one-hot encoding of its atom name ($C_\alpha$, $C_\beta$, N, O, etc.). We consider all backbone and side-chain atoms. On the ligand side, the atom features $\boldsymbol{A}$ are learned by a message passing network (MPN) [42] based on a ligand molecular graph. The MPN and energy function are optimized jointly during training.

**Antibodies**. When the ligand is an antibody, the input to our model is an antibody-antigen binding interface. The binding interface is composed of residues in the antibody complementarity-determining region (CDR) of both heavy and light chains and top 50 antigen residues (epitope) closest to the CDR. The model architecture is depicted in Figure 1b. Different from the small molecule case, we only consider $C_\alpha$ atoms of each antibody/antigen residue. Each residue is represented by a 2560 dimensional pre-trained ESM-2 embedding [18]. For computational efficiency, we freeze ESM-2 weights during training.

### 3.2 Training EBMs with Denoising Score Matching

Given that our training set does not have binding affinity labels, we need to design an unsupervised training objective different from a supervised regression loss. Our key hypothesis is that we can infer the true binding energy function (up to affine equivalence) by maximizing the likelihood of crystal

**Algorithm 1** Training procedure (single data point)

**Require:** A training complex $(\boldsymbol{A}, \boldsymbol{X})$.
 1: Sample a noise level $\sigma$.
 2: Sample rotation vector $\boldsymbol{\omega} \sim \mathcal{N}_{SO(3)}$ with variance $\sigma^2$
 3: Sample translation vector $\boldsymbol{t} \sim \mathcal{N}(0, \sigma^2 \boldsymbol{I})$.
 4: Perturb the coordinates $\tilde{\boldsymbol{X}}$ by applying rigid transformation $(\boldsymbol{\omega}, \boldsymbol{t})$ to the original complex.
 5: Compute the score of energy function $(\tilde{\boldsymbol{\omega}}, \tilde{\boldsymbol{t}})$ based on its gradient (force) $-\partial E/\partial \tilde{\boldsymbol{X}}$.
 6: Minimize DSM objective $\ell_{\mathrm{dsm}}$.

structures in our training set. The motivation of our hypothesis is that a crystal structure is the lowest energy state of a protein-ligand complex. The maximum likelihood objective seeks to minimize the energy of crystal structures since the likelihood of a complex is $p(\boldsymbol{A}, \boldsymbol{X}) \propto \exp(-E(\boldsymbol{A}, \boldsymbol{X}))$.

While maximum likelihood estimation (MLE) is difficult for EBMs due to marginalization, recent works [33, 34] has successfully trained EBMs using denoising score matching (DSM) and proved that DSM is a good approximation of MLE. In standard DSM, we create a perturbed complex by adding Gaussian noise to ligand atom coordinates, i.e., $\tilde{\boldsymbol{X}} = \boldsymbol{X} + \boldsymbol{\epsilon}$, where $\boldsymbol{\epsilon} \sim p(\boldsymbol{\epsilon}) = \mathcal{N}(0, \sigma^2 \boldsymbol{I})$. DSM objective tries to match the score of our model $-\partial E/\partial \tilde{\boldsymbol{X}}$ and the score of the noise distribution $\nabla_{\boldsymbol{\epsilon}} \log p(\boldsymbol{\epsilon}) = -\boldsymbol{\epsilon}/\sigma^2$:

$$\ell_{\mathrm{g}} = \mathbb{E}\big[\|\partial E(\boldsymbol{A}, \tilde{\boldsymbol{X}})/\partial \tilde{\boldsymbol{X}} - \boldsymbol{\epsilon}/\sigma^2\|^2\big] \tag{2}$$

Intuitively, $\ell_{\mathrm{g}}$ forces the gradient to be zero when the input complex is a crystal structure ($\boldsymbol{\epsilon} = 0$). As a result, a crystal structure pose will be at the local minima of an EBM under the DSM objective.

### 3.3 Perturbing a Complex with Rigid Transformation Noise

Nevertheless, adding Gaussian noise is not ideal for protein-ligand binding because it may create nonsensical conformations that violate physical constraints (e.g., an aromatic ring must be planar). A better solution is to create a perturbed complex $(\boldsymbol{A}, \tilde{\boldsymbol{X}})$ via random ligand rotation and translation, similar to molecular docking. To construct a random rotation, we sample a rotation vector $\boldsymbol{\omega}$ from $\mathcal{N}_{SO(3)}$, an isotropic Gaussian distribution over $SO(3)$ rotation group [15] with variance $\sigma^2$. Each $\boldsymbol{\omega} \sim \mathcal{N}_{SO(3)}$ has the form $\boldsymbol{\omega} = \theta \hat{\boldsymbol{\omega}}$, where $\hat{\boldsymbol{\omega}}$ is a vector sampled uniformly from a unit sphere and $\theta \in [0, \pi]$ is a rotation angle with density

$$f(\theta) = \frac{1 - \cos\theta}{\pi} \sum_{l=0}^{\infty} (2l+1) e^{-l(l+1)\sigma^2} \frac{\sin((l+1/2)\theta)}{\sin(\theta/2)} \tag{3}$$

Likewise, we sample a random translation vector $\boldsymbol{t}$ from a normal distribution $\boldsymbol{t} \sim \mathcal{N}(0, \sigma^2 \boldsymbol{I})$. Finally, we apply this rigid transformation to the ligand and compute its perturbed coordinates $\tilde{\boldsymbol{X}} = \boldsymbol{R}_{\boldsymbol{\omega}} \boldsymbol{X} + \boldsymbol{t}$, where $\boldsymbol{R}_{\boldsymbol{\omega}}$ is the rotation matrix given by the rotation vector $\boldsymbol{\omega} = (\boldsymbol{\omega}_x, \boldsymbol{\omega}_y, \boldsymbol{\omega}_z)$.

$$\boldsymbol{R}_{\boldsymbol{\omega}} = \exp(\boldsymbol{W}_{\boldsymbol{\omega}}), \ \boldsymbol{W}_{\boldsymbol{\omega}} = \begin{pmatrix} 0 & -\boldsymbol{\omega}_z & \boldsymbol{\omega}_y \\ \boldsymbol{\omega}_z & 0 & -\boldsymbol{\omega}_x \\ -\boldsymbol{\omega}_y & \boldsymbol{\omega}_x & 0 \end{pmatrix}. \tag{4}$$

Here $\exp$ means matrix exponentiation and $\boldsymbol{W}_{\boldsymbol{\omega}}$ is an infinitesimal rotation matrix. Since $\boldsymbol{W}_{\boldsymbol{\omega}}$ is a skew symmetric matrix, its matrix exponential has the following closed form

$$\boldsymbol{R}_{\boldsymbol{\omega}} = \exp(\boldsymbol{W}_{\boldsymbol{\omega}}) = \boldsymbol{I} + c_1 \boldsymbol{W}_{\boldsymbol{\omega}} + c_2 \boldsymbol{W}_{\boldsymbol{\omega}}^2 \tag{5}$$

$$c_1 = \frac{\sin \|\boldsymbol{\omega}\|}{\|\boldsymbol{\omega}\|}, \quad c_2 = \frac{1 - \cos \|\boldsymbol{\omega}\|}{\|\boldsymbol{\omega}\|^2} \tag{6}$$

Moreover, we do not need to explicitly compute the matrix exponential $\boldsymbol{R}_{\boldsymbol{\omega}}$ since $\boldsymbol{W}_{\boldsymbol{\omega}}$ is the linear mapping of cross product, i.e. $\boldsymbol{\omega} \times \boldsymbol{r} = \boldsymbol{W}_{\boldsymbol{\omega}} \boldsymbol{r}$. Therefore, applying a rotation matrix only involves cross product operations that are very efficient:

$$\boldsymbol{R}_{\boldsymbol{\omega}} \boldsymbol{x}_i = \boldsymbol{x}_i + c_1 \boldsymbol{\omega} \times \boldsymbol{x}_i + c_2 \boldsymbol{\omega} \times (\boldsymbol{\omega} \times \boldsymbol{x}_i) \tag{7}$$

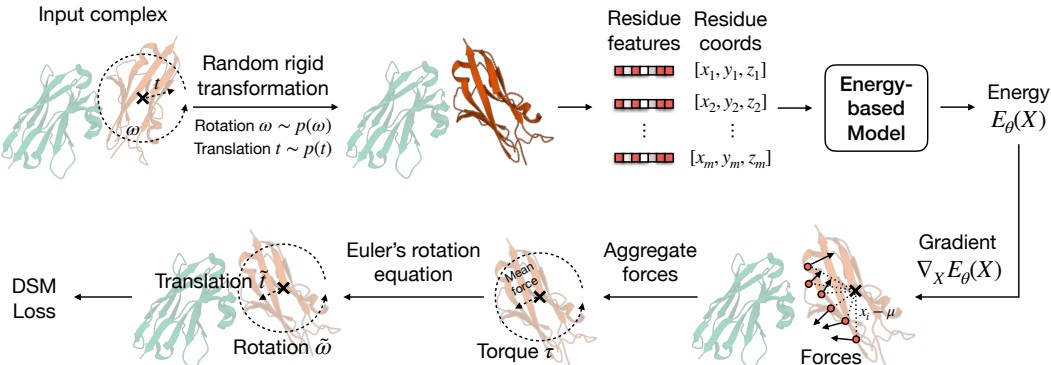

Figure 2: NERE predicts a rotation in three steps. It first calculates pairwise residue force based on the gradient $\boldsymbol{f}_i = (-\partial E/\partial \boldsymbol{x})^\top$. It then predicts the corresponding torque and angular velocity by solving Euler's rotation equation. Lastly, it converts the angular velocity vector to a rotation via matrix exponential map.

### 3.4 Training EBMs with SE(3) Denoising Score Matching

Under this perturbation scheme, DSM aims to match the model score $-\partial E/\partial \tilde{X}$ and the score of rotation and translation noise $\nabla_{\boldsymbol{\omega}} \log p(\boldsymbol{\omega}), \nabla_{\boldsymbol{t}} \log p(\boldsymbol{t})$. Our SE(3) DSM objective is a sum of two losses: $\ell_{\mathrm{dsm}} = \ell_t + \ell_r$, where $\ell_t$ and $\ell_r$ correspond to a translation and a rotation DSM loss. The translation part is straightforward since $\boldsymbol{t}$ follows a normal distribution and $\nabla_{\boldsymbol{t}} \log p(\boldsymbol{t}) = -\boldsymbol{t}/\sigma^2$:

$$\ell_t = \mathbb{E}\big[\|\tilde{\boldsymbol{t}} - \nabla_{\boldsymbol{t}} \log p(\boldsymbol{t})\|^2\big], \qquad \tilde{\boldsymbol{t}} = -\frac{1}{n}\sum_i \partial E/\partial \tilde{\boldsymbol{x}}_i \tag{8}$$

The rotation part of DSM is more complicated. As $\hat{\boldsymbol{\omega}}$ is sampled from a uniform distribution over a sphere (whose density is constant), the density and score of $p(\boldsymbol{\omega})$ is

$$p(\boldsymbol{\omega}) \propto f(\theta), \quad \nabla_{\boldsymbol{\omega}} \log p(\boldsymbol{\omega}) = \nabla_\theta \log f(\theta) \cdot \hat{\boldsymbol{\omega}} \tag{9}$$

In practice, we calculate the density and score by precomputing truncated infinite series in $f(\theta)$. However, the main challenge is that the model score $-\partial E/\partial \tilde{X} \in \mathbb{R}^{n \times 3}$ is defined over atom coordinates, which is not directly comparable with $\nabla_{\boldsymbol{\omega}} \log p(\boldsymbol{\omega}) \in \mathbb{R}^3$ as they have different dimensions. To address this issue, we need to map $-\partial E/\partial \tilde{X}$ to a rotation vector $\tilde{\boldsymbol{\omega}}$ via a rotation predictor $F$ and perform score matching in the rotation space:

$$\ell_r = \mathbb{E}\big[\|\tilde{\boldsymbol{\omega}} - \nabla_{\boldsymbol{\omega}} \log p(\boldsymbol{\omega})\|^2\big], \qquad \tilde{\boldsymbol{\omega}} = F(-\partial E/\partial \tilde{X}) \tag{10}$$

The rotation predictor $F$ is based on Neural Euler's Rotation Equation, which will be described in the next section. The overall training procedure is summarized in Algorithm 1 and Figure 2.

## 4 Neural Euler's Rotation Equation

In this section, we present Neural Euler's Rotation Equation (NERE), which can be used in SE(3) DSM to infer rotational scores from the gradient $-\partial E/\partial \tilde{X}$. To motivate our method, let us first review some basic concepts related to Euler's Rotation Equation.

### 4.1 Euler's Rotation Equations

In classical mechanics, Euler's rotation equation is a first-order ordinary differential equation that describes the rotation of a rigid body. Suppose a ligand rotates around its center mass $\boldsymbol{\mu}$ with angular velocity $\boldsymbol{\omega}$. Euler's rotation equation in an inertial reference frame is defined as

$$\boldsymbol{I}_N \frac{d\boldsymbol{\omega}}{dt} = \boldsymbol{\tau}, \qquad \boldsymbol{\tau} = \sum_i (\boldsymbol{x}_i - \boldsymbol{\mu}) \times \boldsymbol{f}_i \tag{11}$$

where $\boldsymbol{I}_N \in \mathbb{R}^{3 \times 3}$ is the inertia matrix of a ligand, $\boldsymbol{\tau}$ is the torque it received, and $\boldsymbol{f}_i$ is the force applied to a ligand atom $i$. The inertia matrix describes the mass distribution of a ligand and the

torque needed for a desired angular acceleration, which is defined as

$$\boldsymbol{I}_N = \sum_i \|\boldsymbol{x}_i - \boldsymbol{\mu}\|^2 \boldsymbol{I} - (\boldsymbol{x}_i - \boldsymbol{\mu})(\boldsymbol{x}_i - \boldsymbol{\mu})^\top \tag{12}$$

For a short period of time $\Delta t$, we can approximate the new angular velocity $\boldsymbol{\omega}_{t=\Delta t}$ by

$$\frac{d\boldsymbol{\omega}}{dt} \approx \frac{\boldsymbol{\omega}_{t=\Delta t} - \boldsymbol{\omega}_{t=0}}{\Delta t} = \boldsymbol{I}_N^{-1} \tau \tag{13}$$

Since we assume the system is in an inertial reference frame ($\boldsymbol{\omega}_{t=0} = 0$), we have $\boldsymbol{\omega}_{t=\Delta t} = \boldsymbol{I}_N^{-1} \tau \Delta t$ (we set $\Delta t = 0.1$). We note that calculating the inverse $\boldsymbol{I}_N^{-1}$ is cheap because it is a $3 \times 3$ matrix.

## 4.2 Applying NERE to SE(3) DSM

In SE(3) DSM, our goal is to map the gradient $-\partial E / \partial \boldsymbol{x}_i$ to a rotation vector $\boldsymbol{\omega}$. In physics, the gradient $-\partial E / \partial \boldsymbol{x}_i$ is the force $\boldsymbol{f}_i$ of atom $i$. Therefore, the rotation predictor $F$ in Eq.(10) is a simple application of the Euler's Rotation Equation

$$\boldsymbol{\omega} = F(-\partial E / \partial \boldsymbol{X}) = \boldsymbol{I}_N^{-1} \boldsymbol{\tau} \Delta t, \qquad \boldsymbol{\tau} = \sum_i (\boldsymbol{x}_i - \boldsymbol{\mu}) \times (-\partial E / \partial \boldsymbol{x}_i) \tag{14}$$

A nice property of NERE is that it is equivariant under SO(3) rotation group because it is derived from physics. We formally state this proposition as follows and its proof in the appendix.

**Proposition 1.** *Suppose we rotate a ligand so that its new coordinates become $\boldsymbol{x}_i' = \boldsymbol{R}\boldsymbol{x}_i$. The new force $\boldsymbol{f}'$, torque $\boldsymbol{\tau}'$, inertia matrix $\boldsymbol{I}_N'$, and angular velocity $\boldsymbol{\omega}'$ for the rotated complex are*

$$\boldsymbol{f}_i' = \boldsymbol{R}\boldsymbol{f}_i, \boldsymbol{\tau}' = \boldsymbol{R}\boldsymbol{\tau}, \boldsymbol{I}_N' = \boldsymbol{R}\boldsymbol{I}_N \boldsymbol{R}^\top, \boldsymbol{\omega}' = \boldsymbol{R}\boldsymbol{\omega}$$

*In other words, NERE is equivariant under SO(3) rotation group.*

Once we establish SO(3) equivariance, it is easy to satisfy SE(3) equivariance by first placing the rotation center at the origin ($\boldsymbol{x}_i \leftarrow \boldsymbol{x}_i - \boldsymbol{\mu}$), applying the predicted rotation via NERE, and then adding $\boldsymbol{\mu}$ back to each atom.

# 5 Experiments

We evaluate our model on two drug discovery applications: protein-ligand binding and antibody-antigen binding affinity prediction. The experimental setup is described as follows.

## 5.1 Protein-Ligand Binding

**Data**. Our training data has 5237 protein-ligand complexes from the refined subset of PDBbind v2020 database [36]. Our test set has 285 complexes from the PDBbind core set with binding affinity labels converted into log scale. Our validation set has 357 complexes randomly sampled from PDBbind by Stärk et al. [35] after excluding all test cases. The final training set has 4806 complexes (without binding affinity labels) after removing all ligands overlapping with the test set.

**Metric**. We report the Pearson correlation coefficient between true binding affinity and predicted energy $E(\boldsymbol{A}, \boldsymbol{X})$. We do not report root mean square error (RMSE) because our model does not predict absolute affinity values. In fact, shifting $E(\boldsymbol{A}, \boldsymbol{X})$ by any constant will be equally optimal under the DSM objective. We run our model with five different random seeds and report their average.

**Baselines**. We consider three sets of baselines for comparison:

- **Physics-based potentials** calculate binding affinity based on energy functions designed by experts. We consider four popular methods: Glide [8], AutoDock$_{\text{vina}}$ [7], DrugScore$_{2018}$ [6], and MM/GBSA [21]. Among these methods, MM/GBSA is the most accurate but computationally expensive. It takes one hour to calculate energy for just one complex on a 64-core CPU server.
- **Unsupervised models**. Since unsupervised learning is relatively underexplored in this area, we implement two unsupervised EBMs using the same encoder architecture as NERE but different training objectives. The first baseline is the standard Gaussian DSM (see Section 3.2). The second baseline is contrastive learning [4]. For each crystal structure $(\boldsymbol{A}, \boldsymbol{X})$, we apply $K$ random rigid transformations to obtain $K$ perturbed protein-ligand complexes $\boldsymbol{X}_1, \cdots, \boldsymbol{X}_K$ as negative samples. Suppose $-E(\boldsymbol{A}, \boldsymbol{X}_i)$ is the predicted energy for $\boldsymbol{X}_i$, we train our EBM to maximize the likelihood of the crystal structure $\exp(-E(\boldsymbol{A}, \boldsymbol{X})) / \sum_i \exp(-E(\boldsymbol{A}, \boldsymbol{X}_i))$.

| | Sup | Crystal | Docked | | Sup | Crystal | Docked |
|---|---|---|---|---|---|---|---|
| IGN | ✓ | **0.837** | 0.780 | $FANN_{ab}$ | ✓ | $0.325_{.014}$ | $0.326_{.019}$ |
| TankBind | ✓ | n/a | **0.824** | $FANN_{transfer}$ | ✓ | $\mathbf{0.350}_{.033}$ | $\mathbf{0.359}_{.039}$ |
| PLANET | ✓ | 0.811 | 0.799 | AP_PISA | × | 0.323 | 0.144 |
| Glide | × | 0.467 | 0.478 | ZRANK | × | 0.318 | 0.163 |
| $Autodock_{vina}$ | × | 0.604 | 0.578 | PYDOCK | × | 0.248 | 0.164 |
| $DrugScore_{2018}$ | × | 0.602 | 0.540 | ESM-1v | × | 0.038 | 0.038 |
| MM/GBSA | × | 0.647 | 0.629 | ESM-IF | × | 0.024 | 0.025 |
| Contrastive | × | $0.625_{.002}$ | $0.623_{.002}$ | Contrastive | × | $0.308_{.037}$ | $0.304_{.021}$ |
| Gauss DSM | × | $0.638_{.017}$ | $0.632_{.016}$ | Gauss DSM | × | $0.335_{.038}$ | $0.330_{.022}$ |
| NERE DSM | × | $\mathbf{0.656}_{.012}$ | $\mathbf{0.651}_{.011}$ | NERE DSM | × | $\mathbf{0.361}_{.051}$ | $\mathbf{0.360}_{.051}$ |

Table 1: Pearson correlation on PDBBind (left) and SAbDab test sets (right). We report standard deviations for all models implemented in this paper. "Sup" indicates whether a method is supervised or unsupervised. "Crystal"/"Docked" means a model is evaluated on crystal or docked structures. The performance of TankBind on crystal structures is not available because it performs docking and affinity prediction simultaneously in one model.

- **Supervised models**. Most of the existing deep learning models for binding affinity prediction belong to this category. They are typically trained on the entire PDBBind database with over 19000 binding affinity data points. The purpose of this comparison is to understand the gap between supervised and unsupervised models when there is abundant labeled data. For this purpose, we include three top-performing methods (IGN [13], TankBind [19], and PLANET [43]) taken from a recent survey by Zhang et al. (2023).

**Results (crystal structures)**. In our first experiment, we evaluate all methods on the crystal structure of protein-ligand complexes. As shown in Table 1 (left, crystal column), NERE outperforms all physics-based and unsupervised models. We note that NERE is orders of magnitude faster than the second best method MM/GBSA (100ms v.s. 1hr per complex). As expected, the three supervised models perform better than NERE because they are trained on 19000 labeled affinity data points. Nonetheless, NERE recovers almost 80% of their performance without using any labeled data, which implies that binding affinity is closely related to the geometric structure of an input complex.

**Results (docked structures)**. The previous evaluation considers an idealistic setting because crystal structures are not available in real-world virtual screening projects. To this end, we use AutoDock Vina [7] to dock each ligand in the test set to its binding pocket and evaluate all methods on docked protein-ligand complexes. As shown in Table 1 (left, docked column), NERE consistently outperforms the baselines in this setting. Its performance is quite close to the crystal structure setting because docking error is relatively low (median test RMSD=3.55). In summary, these results suggest that NERE has learned an accurate binding energy function useful for structure-based virtual screening.

## 5.2 Antibody-Antigen Binding

**Data**. Our training and test data come from the Structural Antibody Database (SAbDab) [31], which contains 4883 non-redundant antibody-antigen complexes. Our test set is a subset of SAbDab (566 complexes) that has binding affinity labels. Our training set has 3416 complexes (without binding affinity labels) after removing antigen/antibody sequences that appear in the test set. Our validation set comes from Myung et al. [23], which has 116 complexes after removing antibodies or antigens overlapping with the test set.

**Baselines**. Similar to the previous section, we consider three sets of baselines for comparison:

- **Physics-based models**. We consider eight biophysical potentials implemented in the CCharPPI webserver [22] and report the top three models (full results are listed in the appendix).
- **Unsupervised models**. Besides contrastive learning and Gaussian DSM baselines used in Section 5.1, we consider two state-of-the-art protein language models (ESM-1v [20] and ESM-IF [11]). These two models have been successful in predicting mutation effects or binding affinities for

general proteins and we seek to evaluate their performance for antibodies. The input to ESM-1v is the concatenation of an antibody and an antigen sequence and we take the pseudo log-likelihood of antibody CDR residues as the binding affinity of an antibody-antigen pair. The input to ESM-IF is the crystal structure of an antibody-antigen complex and we take the conditional log-likelihood of antibody CDR residues given the backbone structure as its binding affinity.

- **Supervised models**. We also compare NERE with supervised models trained on external binding affinity data. The model is based on the same frame-averaging neural network (FANN) [28] and pre-trained ESM-2 residue embeddings used by NERE.[1] Since all labeled data from SAbDab are in the validation and test sets, we draw additional data from an protein-protein binding mutation database (SKEMPI) [12]. We obtain 5427 binding affinity data points after removing all complexes appeared in the test set, from which 273 instances are antibody-antigen complexes. We explore two training strategies: 1) training on 273 antibody-antigen data points only (FANN$_{ab}$); 2) training on 5427 data points first and then finetuning on 273 antibody-antigen data points (FANN$_{transfer}$).

**Results (Crystal structure)**. We first evaluate all models on crystal structures and report the Pearson correlation between true and predicted binding energy. As reported in Table 1 (right, crystal column), NERE significantly outperforms physics-based methods and protein language models. ESM-1v and ESM-IF have almost zero correlation because they only model the likelihood of the antibody sequence. ESM-IF models the conditional likelihood of a sequence given its structure while NERE models the likelihood of the entire protein complex.It is crucial to model the likelihood of structures because binding affinity depends on the relative orientation between an antibody and an antigen. We also find that NERE outperforms the supervised models, even with a small gain over FANN$_{transfer}$ trained on 5427 labeled data points. Overall, these results highlight the advantage of unsupervised learning in low-data regimes.

**Results (Docked structure)**. In our second experiment, we evaluate all methods on docked complexes to emulate a more realistic scenario. We use the ZDOCK program [26] to predict the structure of all antibody-antigen complexes in the test set (its median RMSD is 19.4). As reported in Table 1 (right, docked column), our model still outperforms all the baselines in this challenging setting. The performance of EBMs are quite robust to docking errors regardless of the training algorithm (contrastive learning, Gaussian DSM, or NERE DSM). In summary, our model is capable to predict antibody-antigen binding even when crystal structure is not available.

### 5.3 Ablation Studies and Visualization

**Visualizing energy landscape**. First, we study how the learned energy changes with respect to ligand orientations. Given an input complex, we perform a grid search of ligand rotation angles $\boldsymbol{\omega} = [\boldsymbol{\omega}_1, \boldsymbol{\omega}_2, \boldsymbol{\omega}_3]$ and plot the predicted energy for each pose. As 3D contour plot is hard to visualize, we decompose it into three 2D contour plots by fixing one of the three axis ($\boldsymbol{\omega}_1, \boldsymbol{\omega}_2, \boldsymbol{\omega}_3$) to zero. Ideally, the crystal structure ($\boldsymbol{\omega} = [0, 0, 0]$) should be the local minimum because it is physically the most stable conformation. Figure 3a-b show contour plots for small molecules and antibody-antigen complexes. We find that their crystal structures are located relatively near the local minimum.

**Visualizing interaction energy**. Since the predicted energy is a summation of all pairwise interactions $\sum_{i,j:d_{i,j}<20\text{Å}} \phi_o(\boldsymbol{h}_i, \boldsymbol{h}_j)$, we seek to visualize the contribution of different residues to predicted binding energy. Figure 3c is one example. Each row and column in this heatmap represent an epitope residue and an antibody CDR residue, respectively. Each entry in the heat map is the interaction energy between two residue $\phi_o(\boldsymbol{h}_i, \boldsymbol{h}_j)$. An entry is left blank if the distance $d_{i,j} > 20\text{Å}$. Interestingly, we find that the model pays the most attention to CDR-H3 and CDR-L3 residues. In most test cases, their energy is much lower than other CDR residues. This agrees with the domain knowledge that CDR-H3 and CDR-L3 residues are the major determinant of binding.

**Ablation studies**. Lastly, we perform four ablation studies to understand the importance of different model components. We first replace our SE(3)-invariant protein encoder with a non-invariant neural network where atom 3D coordinates are directly concatenated to atom features in the input layer. As shown in Figure 3d, the model performance drops significantly when the encoder is not SE(3) invariant. In addition, we run NERE DSM by removing the rotation $\ell_r$ or translation DSM term $\ell_t$ from the objective $\ell_{\text{DSM}} = \ell_t + \ell_r$. We find that removing either of them substantially hurts the model

---

[1]We have also tried other neural network architectures like 3D convolutional neural networks and graph convolutional networks, but they did not achieve better performance. Their results are shown in the appendix.

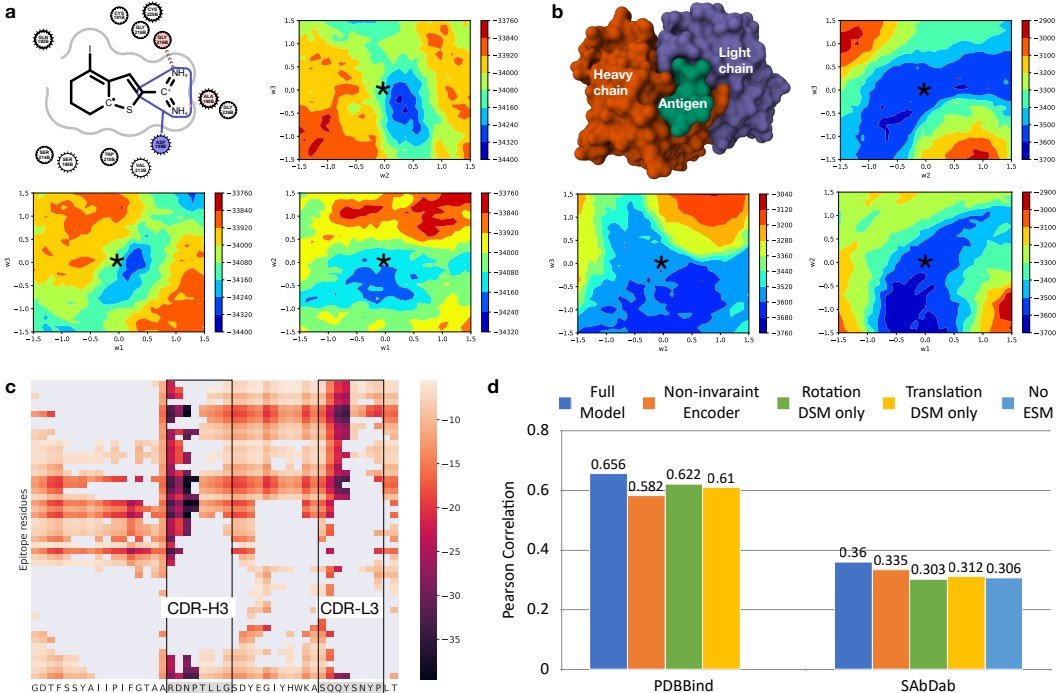

Figure 3: a-b) Visualizing learned energy landscape for small molecules and antibodies. We perform a grid search of ligand rotation angles $\boldsymbol{\omega} = [\boldsymbol{\omega}_1, \boldsymbol{\omega}_2, \boldsymbol{\omega}_3]$ and plot the predicted energy as 2D contour plots with one of the axis $\boldsymbol{\omega}_i$ fixed. The crystal structure is at the origin (marked with *) and supposed to be the local minimum of the energy landscape. c) The heat map of our learned energy function. Each entry represent the binding energy between residues $(i, j)$ (the darker the stronger). An entry is left blank (grey) if their distance $d_{ij} > 20\text{Å}$. Our model correctly puts more attention to CDR-H3/L3 residues. d) Ablation studies on different model components.

performance. Therefore, it is crucial to consider both rotation and translation degree of freedom in unsupervised binding energy prediction. Lastly, we train NERE DSM by replacing ESM-2 amino acid embedding with one-hot encoding (this change only applies to the antibody model). We find that ESM-2 is helpful but NERE DSM is still able to infer binding affinity with reasonable accuracy after removing language model features.

## 6 Discussion

In this paper, we developed an energy-based model for unsupervised binding affinity prediction. The energy-based model was trained under SE(3) denoising score matching where the rotation score was predicted by Neural Euler's Rotation Equation. Our results show that the learned energy correlates with experimental binding affinity and outperforms supervised models in the antibody case.

**Limitations**. Indeed, there are many ways to improve our model. Our current model for antibodies only considers their backbone structure ($C\alpha$ atoms) and ignores all side-chain atoms. While the model for small molecules include all atoms, we have not incorporated the flexibility of side chains in a protein or rotatable bonds in a ligand. Considering their interaction is crucial for protein-ligand binding prediction. Our future work is to extend our encoder with all-atom structures and our NERE DSM algorithm with side-chain torsional rotations.

**Broader Impacts**. Our method is applicable to a wide range of biological areas like drug discovery, immunology, and structural biology. We believe our work does not have any potential negative societal impacts since our aim is to accelerate scientific discovery.

**Code and Data**. Our code and data are available at `github.com/wengong-jin/DSMBind`.

## Acknowledgements

We would like to thank Divya Nori, Chenyu Wang, and anonymous reviewers for their valuable feedback on the manuscript. Wengong Jin is supported by the BroadIgnite Award and the Eric and Wendy Schmidt Center at the Broad Institute of MIT and Harvard. Caroline Uhler was partially supported by NCCIH/NIH (1DP2AT012345), ONR (N00014-22-1-2116), AstraZeneca, and a Simons Investigator Award.

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

## A NERE Rotational Equivariance

**Proposition 1**. Suppose we rotate a protein-ligand complex so that new coordinates become $\boldsymbol{x}_i' = \boldsymbol{R}\boldsymbol{x}_i$. The new force $\boldsymbol{f}'$, torque $\boldsymbol{\tau}'$, inertia matrix $\boldsymbol{I}_N'$, and angular velocity $\boldsymbol{\omega}'$ under the rotated complex are

$$\boldsymbol{f}_i' = \boldsymbol{R}\boldsymbol{f}_i, \boldsymbol{\tau}' = \boldsymbol{R}\boldsymbol{\tau}, \boldsymbol{I}_N' = \boldsymbol{R}\boldsymbol{I}_N\boldsymbol{R}^\top, \boldsymbol{\omega}' = \boldsymbol{R}\boldsymbol{\omega},$$

*Proof.* After rotating the whole complex, the energy function $E(\boldsymbol{A}, \boldsymbol{X}') = E(\boldsymbol{A}, \boldsymbol{X})$ since the encoder is SE(3)-invariant. Given that $\boldsymbol{x}_i = \boldsymbol{R}^\top\boldsymbol{x}_i'$ and $\partial\boldsymbol{x}_i/\partial\boldsymbol{x}_i' = \boldsymbol{R}^\top$, the new force becomes

$$\boldsymbol{f}_i' = \left(\frac{\partial E(\boldsymbol{A}, \boldsymbol{X}')}{\partial\boldsymbol{x}_i'}\right)^\top = \left(\frac{\partial E(\boldsymbol{A}, \boldsymbol{X})}{\partial\boldsymbol{x}_i}\frac{\partial\boldsymbol{x}_i}{\partial\boldsymbol{x}_i'}\right)^\top = \left(\frac{\partial\boldsymbol{x}_i}{\partial\boldsymbol{x}_i'}\right)^\top\boldsymbol{f}_i = \boldsymbol{R}\boldsymbol{f}_i$$

Based on the definition of torque and the fact that cross products satisfy $\boldsymbol{R}\boldsymbol{x} \times \boldsymbol{R}\boldsymbol{y} = \boldsymbol{R}(\boldsymbol{x} \times \boldsymbol{y})$, we have

$$\boldsymbol{\tau}' = \sum_i(\boldsymbol{x}_i' - \boldsymbol{\mu}') \times \boldsymbol{f}_i' = \sum_i(\boldsymbol{R}\boldsymbol{x}_i - \boldsymbol{R}\boldsymbol{\mu}) \times \boldsymbol{R}\boldsymbol{f}_i$$

$$= \boldsymbol{R}\sum_i(\boldsymbol{x}_i - \boldsymbol{\mu}) \times \boldsymbol{f}_i = \boldsymbol{R}\boldsymbol{\tau}$$

Likewise, using the fact that $\boldsymbol{R}\boldsymbol{R}^\top = \boldsymbol{I}$, the new inertia matrix becomes

$$\boldsymbol{I}_N' = \sum_i\|\boldsymbol{x}_i' - \boldsymbol{\mu}'\|^2\boldsymbol{I} - (\boldsymbol{x}_i' - \boldsymbol{\mu}')(\boldsymbol{x}_i' - \boldsymbol{\mu}')^\top$$

$$= \sum_i\|\boldsymbol{x}_i - \boldsymbol{\mu}\|^2\boldsymbol{I} - (\boldsymbol{R}\boldsymbol{x}_i - \boldsymbol{R}\boldsymbol{\mu})(\boldsymbol{R}\boldsymbol{x}_i - \boldsymbol{R}\boldsymbol{\mu})^\top$$

$$= \sum_i\boldsymbol{R}\|\boldsymbol{x}_i - \boldsymbol{\mu}\|^2\boldsymbol{I}\boldsymbol{R}^\top - \boldsymbol{R}(\boldsymbol{x}_i - \boldsymbol{\mu})(\boldsymbol{x}_i - \boldsymbol{\mu})^\top\boldsymbol{R}^\top$$

$$= \boldsymbol{R}\left(\sum_i\|\boldsymbol{x}_i - \boldsymbol{\mu}\|^2\boldsymbol{I} - (\boldsymbol{x}_i - \boldsymbol{\mu})(\boldsymbol{x}_i - \boldsymbol{\mu})^\top\right)\boldsymbol{R}^\top$$

$$= \boldsymbol{R}\boldsymbol{I}_N\boldsymbol{R}^\top$$

For angular velocity, we have

$$\boldsymbol{\omega}' = C\boldsymbol{I}_N'^{-1}\boldsymbol{\tau}' = C\boldsymbol{R}\boldsymbol{I}_N^{-1}\boldsymbol{R}^\top\boldsymbol{R}\boldsymbol{\tau} = C\boldsymbol{R}\boldsymbol{I}_N^{-1}\boldsymbol{\tau} = \boldsymbol{R}\boldsymbol{\omega}$$

Therefore, NERE layer is equivariant under rotation. $\square$

## B Experimental Details

**Protein encoder architecture**. The energy function needs to be SE(3)-invariant and differentiable with respect to $\boldsymbol{X}$. Thus, we adopt the frame averaging neural network [28] so that $E$ directly takes coordinates $\boldsymbol{X}$ as input rather than a distance matrix. To be specific, our energy function is parameterized as follows

$$\{\boldsymbol{h}_1, \cdots, \boldsymbol{h}_n\} = \frac{1}{|\mathcal{G}|}\sum_{g_k \in \mathcal{G}}\phi_h(\boldsymbol{A}, g_k(\boldsymbol{X})) \tag{15}$$

The encoder $\phi_h$ is a modified transformer network [17] that learns atom representations $\boldsymbol{h}_1, \cdots, \boldsymbol{h}_n$ based on atom features $\boldsymbol{A}$ and coordinates $\boldsymbol{X}$. Specifically, the model first projects the coordinates $\boldsymbol{x}_i$ onto a set of eight frames $\{g_k(\boldsymbol{x}_i)\}$ defined in Puny et al. [28], concatenate the projected coordinates with atom features $\boldsymbol{a}_i$, encode the vector sequence $[\boldsymbol{a}_1, g_k(\boldsymbol{x}_1)], \cdots [\boldsymbol{a}_n, g_k(\boldsymbol{x}_n)]$ to their hidden representations $\boldsymbol{h}_1^k, \cdots, \boldsymbol{h}_n^k$, and then average the frame representations $\boldsymbol{h}_i = \sum_k \boldsymbol{h}_i^k/8$ to maintain SE(3) invariance.

**Model hyperparameters**. In the small molecule case, our model has two components: molecular graph encoder (MPN) and frame-averaging encoder. For the MPN encoder, we use the default

|            | Crystal        | Docked         |
|------------|----------------|----------------|
| ZRANK      | 0.318          | 0.163          |
| RosettaDOCK| 0.064          | 0.025          |
| PYDOCK     | 0.248          | 0.164          |
| SIPPER     | -0.138         | 0.003          |
| AP_PISA    | 0.323          | 0.144          |
| FIREDOCK   | 0.101          | -0.052         |
| FIREDOCK_AB| 0.199          | 0.042          |
| CP_PIE     | 0.234          | 0.120          |
| 3D CNN     | $0.286_{.021}$ | $0.162_{.017}$ |
| GNN        | $0.244_{.075}$ | $0.249_{.075}$ |
| NERE (ours)| $\mathbf{0.361}_{.051}$ | $\mathbf{0.360}_{.051}$ |

Table 2: Additional baselines on the SAbDab test set.

hyperparameter from Yang et al. [42]. For the protein encoder, we set hidden layer dimension to be 256 and try encoder depth $L \in \{1, 2, 3\}$ and distance threshold $d \in \{5.0, 10.0\}$. In the antibody case, we try encoder depth from $L \in \{1, 2, 3\}$ and distance threshold $d \in \{10.0, 20.0\}$. In both cases, we select the hyperparameter that gives the best Pearson correlation on the validation set.

**Docking protocol**. For Autodock Vina, we use its default docking parameters with docking grid dimension of 20Å, grid interval of 0.375Å, and exhaustiveness of 32. For ZDOCK, we mark antibody CDR residues as ligand binding site and generate 2000 poses for each antibody-antigen pair. We re-score those 2000 poses by ZRANK2 and select the best candidate.

**Additional results**. We include additional results for the antibody-antigen binding task. We first compare our method with additional physic-based potentials implemented in the CCharPPI web server [22], including ZRANK [24], RosettaDock [1], PyDock [10], SIPPER [27], AP_PISA [39], CP_PIE [30], FIREDOCK, and FIREDOCK_AB [2]. As shown in Table B, the performance of these models are much lower than NERE.

We have also explored more protein encoder architecture as additional supervised learning baselines. We consider a 3D convolutional neural network (3D CNN) and a graph convolutional network (GCN) implemented in the Atom3D package [38]. For the GCN model, we extend its implementation to include ESM-2 residue embedding. We find that the performance of these two models are much worse than FANN when trained on 273 antibody-antigen complexes from SKEMPI. Therefore, we choose the FANN architecture as our default supervised model architecture for the rest of our analysis.

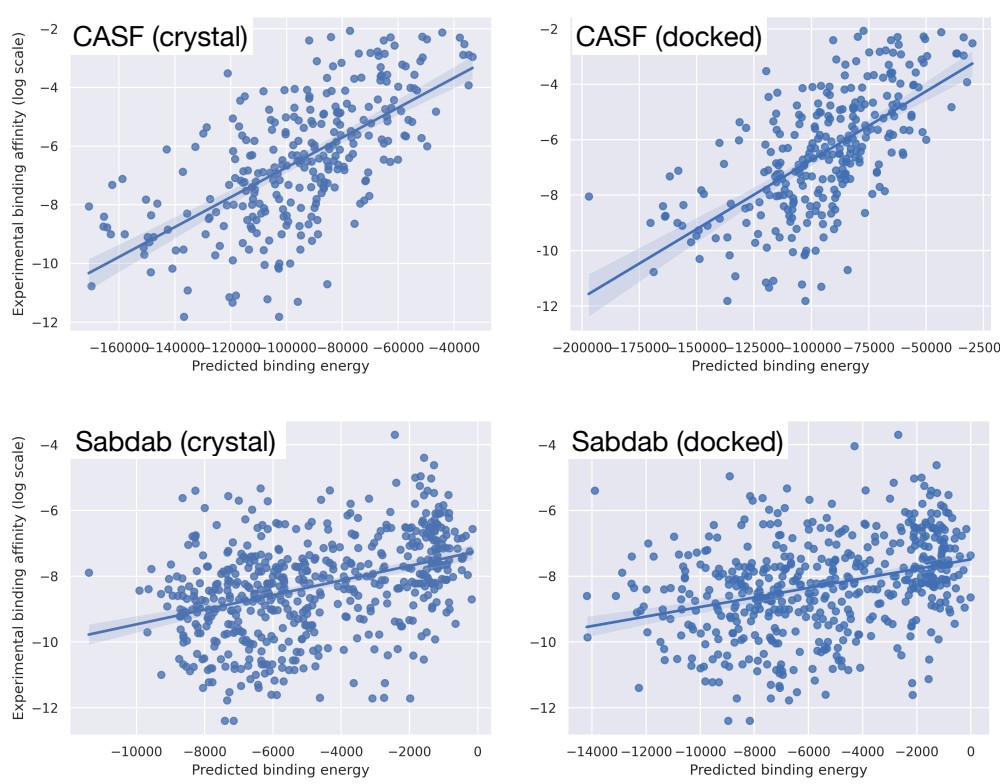

Figure 4: Correlation between predicted and experimental binding energy.

