# OpenReview forum: "Unsupervised Protein-Ligand Binding Energy Prediction via Neural Euler's Rotation Equation"
_NeurIPS.cc/2023/Conference — NeurIPS 2023 poster_

### Official Review · Reviewer_UbeA · 2023-06-21

**Soundness:** 3 good
**Presentation:** 3 good
**Contribution:** 2 fair
**Rating:** 4
**Confidence:** 4

**Summary:**

In this paper, the authors developed an energy-based model for unsupervised binding affinity prediction. The energy-based model was trained under SE(3) denoising score matching where the rotation score was predicted by Neural Euler’s Rotation Equation. Experiments on protein-ligand binding and antibody-antigen binding are conducted.

**Strengths:**

1. The paper is well-written and easy to follow.
2. The proposed method outperforms all unsupervised baselines and supervised baselines in the antibody case.
3. The code and data are provided.
4. Systematic ablation studies are performed to show the effectiveness of modules.



**Weaknesses:**

1. The idea of using unsupervised generative models for binding affinity prediction is not new. The authors fail to discuss or compare with related works [1,2]

[1] Luo et al., Rotamer Density Estimator is an Unsupervised Learner of the Effect of Mutations on Protein-Protein Interaction, ICLR 23
[2] Guan et al., 3D Equivariant Diffusion for Target-Aware Molecule Generation and Affinity Prediction, ICLR 23

2. The authors only calculate the Pearson correlation coefficient. For the binding affinity prediction, other metrics such as RMSE, MAE, AUROC, and Pearson correlation coefficient.

3. The authors fail to consider the flexibility of proteins and ligands. The sidechains of proteins are also ignored. However, these factors are quite important for protein-ligand/protein binding.

**Questions:**

please see the weakness.

**Limitations:**

The limitations are well discussed.

---

> ### Author Rebuttal · Authors · 2023-08-09
>
> Thank you for your insightful comments and suggestions. Please read our response below and let us know if you have more questions.
>
> **Q1**: The authors fail to consider the flexibility of proteins and ligands. The side-chains of proteins are also ignored. However, these factors are quite important for protein-ligand and protein binding.
>
> We agree that it is beneficial to model the flexibility of ligands and incorporate the side-chain information of proteins. Given the limited time, we partially address this issue by including all protein atoms (including side-chains) as input instead of only $C_\alpha$ atoms. This modification only changes the input of our model while the model architecture and NERE DSM objective remains the same. We refer to this extended model as NERE DSM (all-atom). Our results (rebuttal Figure 1e) suggest that including side-chain atoms substantially improves the performance of our method on some of the datasets. Please read our global response above  for details. In the future, we plan to model the flexibility of rotatable bonds and protein side-chain atoms by introducing a side-chain DSM loss, which adds rotation noise on the rotatable bonds and side-chain atoms and use NERE to predict the rotation noise.
>
> **Q2**: The authors only calculate the Pearson correlation coefficient. For the binding affinity prediction, including other metrics such as RMSE, MAE, and Spearman correlation coefficient would be helpful.
>
> We report Spearman correlation and p-values as additional metrics in rebuttal Table 1. Our model (NERE DSM) remains the best under all three metrics. Unfortunately, it was not possible to report root mean square error (RMSE) or mean absolute error (MAE) because our model does not predict absolute affinity values. Shifting or scaling $E(A, X)$ by any constant will be equally optimal under the DSM objective.
>
> **Q3**: The idea of using unsupervised generative models for binding affinity prediction is not new. The authors fail to discuss or compare with related works [1,2].
>
> Thank you for suggesting these papers and they are quite related. Luo et al. [1] proposed a flow-based generative model to estimate the probability distribution of protein side-chain conformations. Their unsupervised model (named RDE-Linear) uses the learned entropy of side-chains to predict $\Delta\Delta G$ of protein mutations. RDE-Linear and NERE DSM are complementary to each other because they consider different degrees of freedom (DOF). RDE-Linear considers the DOF of side-chains and the backbone structure is fixed. NERE DSM considers the DOF of backbone structure (i.e., different docking angles) but not the DOF of side-chains.
>
> To compare NERE DSM with RDE-Linear, we train our model (the all-atom version) on approximately 27000 non-redundant protein-protein complexes (downloaded from PDB) and evaluate it on the entire SKEMPI mutation effect prediction benchmark. For each downloaded complex, we decompose it into pairs of two chains and remove any pairs of chains with buried surface area less than 500. We then clustered these proteins and took one representative complex from each cluster. In each training step, we randomly rotate one of the proteins in a complex and update the model with our NERE DSM objective. The input to our model includes side-chain atoms because the downstream task is mutation effect prediction.
>
> At test time, given a pair of wild type and mutated protein complexes, our model first predicts their binding energy $E_\mathrm{wildtype}$ and $E_\mathrm{mutated}$ and calculates $\Delta\Delta G=E_\mathrm{mutated} - E_\mathrm{wildtype}$. Following Luo et al. [1], we report per-structure and overall Pearson and Spearman correlation between predicted and experimental $\Delta\Delta G$. To compute per-structure correlation, we group mutations by structure, discard groups with less than 10 mutation data points, and calculate correlations for each structure separately. As shown in the table below, we find that NERE DSM achieves a much higher performance than RDE-Linear on three out of four metrics. We visualize the overall correlation between predicted and experimental $\Delta\Delta G$ in rebuttal Figure 1h.
>
> | SKEMPI | Pearson (per structure) | Spearman (per structure) | Pearson (overall) | Spearman (overall) |
> |----|----|---|----|---|
> | RDE-Linear | 0.290 | 0.263 | **0.419** | 0.3514 |
> | NERE DSM (all-atom) | **0.388±0.020**  | **0.402±0.035** | 0.399±0.039 | **0.402±0.013** |
>
> Guan et al. [2] proposed TargetDiff, a diffusion model for 3D small molecule design. Similar to our method, TargetDiff is trained on cocrystal structures of protein-ligand complexes in an unsupervised manner. They demonstrated that the unsupervised features learned by TargetDiff could improve supervised affinity prediction. Specifically, they augmented EGNN [3] with the features learned by TargetDiff and trained the augmented model (EGNN + TargetDiff) on the PDBBind training set. To compare our method with TargetDiff, we followed their training/validation/test split and fine-tuned the pre-trained NERE DSM model on the PDBBind training set. As shown in the table below, our method outperforms the EGNN + TargetDiff baseline, which demonstrates the advantage of our approach. The baseline results are copied from Guan et al. 2023.
>
> | TargetDiff test set  | Pearson | Spearman |
> |----------------------|----------------|---------------|
> | TransCPI | 0.576 | 0.540 |
> | MONN | 0.624 | 0.589 |
> | IGN | 0.698 | 0.641 |
> | EGNN | 0.648 | 0.598 |
> | EGNN + TargetDiff  | 0.680 | 0.637 |
> | NERE DSM (pretrained) | 0.500±0.004 | 0.516±0.013 |
> | NERE DSM (fine-tuned) | **0.703±0.007**  | **0.656±0.017** |
>
> References
> 1. Luo et al., Rotamer Density Estimator is an Unsupervised Learner of the Effect of Mutations on Protein-Protein Interaction, ICLR 23
> 2. Guan et al., 3D Equivariant Diffusion for Target-Aware Molecule Generation and Affinity Prediction, ICLR 23
> 3. Satorras et al., E(n) equivariant graph neural networks. ICML 2021

---

> > ### Comment · Reviewer_UbeA · 2023-08-16
> > **Reply to the authors**
> >
> > I have read the reply and appreciate the author's reply. Thanks!

---

### Official Review · Reviewer_KYrm · 2023-06-30

**Soundness:** 3 good
**Presentation:** 4 excellent
**Contribution:** 3 good
**Rating:** 7
**Confidence:** 3

**Summary:**

The authors propose an energy-based model for unsupervised binding affinity estimation, which is trained by SE(3) denoising score matching (DSM). Different from standard DSM, they add noise by random rotations and translations on the ligand. Utilizing Euler's rotation equations, the rotation matrix can be derived from predicted forces on each atom, making it possible to directly supervise on the predicted forces instead of the rotation matrix. Experiments on protein-ligand binding and antibody-antigen binding demonstrate the superiority of the proposed method over existing unsupervised method.

**Strengths:**

1. Using Euler's rotation equations to formulate the supervision on the rotations as supervision on the predicted atomic forces is novel and clever.
2. Unsupervised learning on binding affinity is important and meaningful as precise affinity labels are hard to obtain in practical.
3. The docked setting is interesting and reals the potential applications of the proposed method in real scenarios where crystal structure is often hard to obtain.
4. It's very interesting to see that pair-wise energies are mostly lower at CDR-H3 and CDR-L3, which perfectly aligns with the domain knowledge.

**Weaknesses:**

1. The learnt probability distribution $p(A, X)$ might not be well aligned with the actual $p(A, X)$ because the actual one is more complicated than the prior distribution of rotation and translation noises. In this sense, for the learnt $p(A, X)$, $p(A, X) \propto \exp(-E(A, X))$ might not hold.

**Questions:**

1. Under the docked setting, the RMSD is quite large for the generated antibody-antigen complexes (median 19.4), however, the correlation is similar to the crystal setting. Does that mean the model actually did not learn useful geometric interaction information? Otherwise, with such badly docked structures, the model should have much worse performance because the structures can not provide accurate geometric information.
2. Will the correlation be improved if you first denoise the input structure (regardless whether it is crystal structure or docked structure) with the model, then output the energy from the denoised structure?

**Limitations:**

The authors have mentioned that the limitation might be only considering the alpha carbon in the antibody-antigen setting. Perturbations on the side chains can be designed with a similar strategy in the future.

---

> ### Author Rebuttal · Authors · 2023-08-09
>
> Thank you for your insightful comments and suggestions. Please read our response below and let us know if you have more questions.
>
> **Q1**: Under the docked setting, the RMSD is quite large for the generated antibody-antigen complexes (median 19.4), however, the correlation is similar to the crystal setting. Does that mean the model actually did not learn useful geometric interaction information?
>
> The reason why our model is robust to docking error is that we use a very loose threshold for residue contact. In NERE DSM, the predicted binding energy is a sum of pairwise residue interaction energy: $E(A, X) = \sum_{i,j: d_{ij} < 20} f_o(h_i, h_j)$. In this formula, we consider two residues to be interacting (a.k.a., forming a contact) if their $C_\alpha$ distance $d_{ij}<20$. This threshold is much larger than standard practice (5-8Å). In fact, we find that the model becomes a lot more sensitive to docking error if we set the contact threshold to 10Å or lower. With a tighter contact threshold, the true contacts are likely to be lost in the docked setting and the model cannot extract useful geometric information for binding energy prediction.
>
> **Q2**: Will the correlation be improved if you first denoise the input structure (regardless whether it is crystal structure or docked structure) with the model, then output the energy from the denoised structure?
>
> We found that the performance is roughly the same with or without this denoising step at inference time (see the table below).
>
> |   | CASF (crystal) | CASF (docked) | SAbDab (crystal) | SAbDab (docked) |
> |----------------------|----------------|---------------|------------------|-----------------|
> | without denoising | 0.656 ± 0.012 | 0.651 ± 0.011 | 0.361 ± 0.051 | 0.360 ± 0.051 |
> | with denoising | 0.655 ± 0.013 | 0.649 ± 0.012 | 0.367 ± 0.029  | 0.359 ± 0.043 |

---

> > ### Comment · Reviewer_KYrm · 2023-08-12
> > **Thanks**
> >
> > Thanks for the response. However, I'm still concerned with the antigen-antibody experiments on docked structures. A median RMSD of 19.4 actually means that the orientation and relative position of the antibody to the antigen are barely correct. Though with larger epitope the correct interaction pairs might still be included in the calculation of the energy, they are not providing the correct geometric information! For example, an actually interacting pair might has long distance in the docked structure, and if the model learnt the true correlation between the geometry and the energy (e.g. interacting pairs with shorter distances might contribute more to the binding affinity), it should fail on such cases because the provided geometric information is wrong! Therefore, could you plot the correlation between the docking RMSD and the error of the predicted energy? It is better if you could compare the pair-wise energy contribution predicted by the model to those calculated by physical softwares (e.g. Rosetta).

---

> > > ### Author Response · Authors · 2023-08-16
> > > **Additional analysis**
> > >
> > > Thank you for your comments. To analyze how docking RMSD affects the performance of NERE DSM, we group the SAbDab test set into multiple bins based on RMSD and report the performance of NERE DSM for each bin. The result is shown in the table below. Here are our main findings:
> > > 1. We realize that 42.8% of the test cases have RMSD less than 10Å. According to CAPRI [1], a docking pose with RMSD less than 10Å is considered correct. While the median RMSD is 19.4Å, we still have a fair number of docked instances that are correct.
> > > 2. We find that the model performance decreases as docking RMSD increases. When RMSD is above 40Å, the performance is almost zero, which shows that the performance of our model is influenced by the docking error.
> > > 3. We notice that the model performance is surprisingly decent when docking RMSD is between 10Å and 30Å. It seems that the learned energy function is not sensitive enough to geometrical changes. We agree that some geometric information seems to be lost in the learned representation, which reveals the limitation of the encoder architecture we used in this paper (frame-averaging neural network [2]). We will explore alternative geometric encoder architectures in the future to improve our method.
> > > 4. We did not compare NERE's pairwise energy contribution with Rosetta because the performance of Rosetta is quite poor on our test set (correlation = 0.025). We were afraid that the Rosetta pairwise energy itself may be inaccurate.
> > >
> > > In summary, our results show that the performance of NERE is influenced by the docking error, but it is not sensitive as we expected. Please let us know if you have any more questions.
> > >
> > > |  RMSD range | # test cases | Correlation |
> > > |------:|-------------:|------------:|
> > > |  0-10 |      238     |    0.506    |
> > > | 10-20 |      47      |    0.441    |
> > > | 20-30 |      69      |    0.410    |
> > > | 30-40 |      59      |    0.268    |
> > > | 40-50 |      51      |    0.030    |
> > > | >50   |      91      |    -0.031   |
> > >
> > > [1] Janin et al., CAPRI: a Critical Assessment of PRedicted Interactions, Proteins 2003
> > >
> > > [2] Puny et al., Frame Averaging for Invariant and Equivariant Network Design, ICLR 2022

---

> > > > ### Comment · Reviewer_KYrm · 2023-08-17
> > > > **Thanks for your response**
> > > >
> > > > Thanks for your careful response! I think the additional analysis has addressed my concern. I will keep recommending the acceptance of the paper.

---

### Official Review · Reviewer_3x52 · 2023-07-04

**Soundness:** 3 good
**Presentation:** 3 good
**Contribution:** 3 good
**Rating:** 6
**Confidence:** 3

**Summary:**

The authors address the problem of protein-ligand binding. Specifically, the authors reformulate binding energy prediction as a generative modeling task: they train an energy-based model on a set of unlabelled protein-ligand complexes using denoising score matching and interpret its log-likelihood as binding affinity. The key contribution of this work is a  new equivariant rotation prediction network for SE(3) DSM called Neural Euler’s Rotation Equations (NERE).

**Strengths:**

* The work is easy to follow and ideas are presented in a clear way
* The proposed approach (with certain particularities) generalizes to both small and large molecules

**Weaknesses:**

NERE DSM relies on cocrystal structures for training and cocrystal/docked structures for binding prediction of a new protein-ligand / Ab-Ag pair, which can be limiting in terms of experimental/computational resources. In addition, NERE DSM does not predict binding affinity directly, hence the model is evaluated using correlation instead of RMSE. The utility of NERE DSM in real-world scenarios where we seek to predict affinities for a large set of protein-ligand / Ab-Ag pairs seems limited. Moreover, the extremely slow rate at which new cocrystal data becomes available is an additional limiting factor. Could the authors comment on this?

**Questions:**

### Questions
* I’m curious, why were only the top 50 antigen residues chosen instead of all residues within a certain distance, as done for the protein in the small molecule case?

* Could you provide more details on the nature of the test splits? Are they IID w.r.t. the training set, or are they split by sequence similarity/some other criteria?

###  Questions regarding protein-ligand binding:
* Why did the authors choose to compare their model to supervised models trained on 19K labeled affinity datapoints? Did I understand correctly that the 4806 datapoints used to train NERE DSM and the 19K are datapoints used to train the supervised models are disjoint? Is there no cases where affinity data is available together with cocrystal data so both models could be trained on the same dataset?

* How long does AutoDock Vina take to dock a protein-ligand pair on average (in your case)?


###  Questions regarding Ab-Ag binding:
* It is not entirely clear to me, are the supervised pre-trained?

* I have the same question here as for protein-ligand binding, is there no cases where affinity data is available together with cocrystal data so both (unsupervised and supervised) models could be trained on the same dataset?

* How long does ZDOCK take to dock a protein-ligand pair on average (in your case)?

###  Questions regarding ablation studies:
* In lines 292-294, the authors say “Interestingly, we find that the model pays the most attention to CDR-H3 and CDR-L3 residues. In most test cases, their energy is much lower than other CDR residues. This agrees with the domain knowledge that CDR-H3 and CDR-L3 residues are the major determinant of binding “. To support this claim, the authors only show one Ab-Ag example which does not guarantee that the same trend is observed across all samples in the test set. Could you report a global metric, for example, the average difference in the test set between energies in CDRs vs energies in remaining Ab regions (plus statistical significance)?

### Minor comments
* In lines 72-73, “Due to scarcity of binding affinity data, very few work has explored supervised learning for antibody binding affinity prediction.” Should be “Due to scarcity of binding affinity data, very few works have explored supervised learning for antibody binding affinity prediction.”
*  Experiments were repeated with 5 random seeds but only the average values were reported in table 1, could you also report the standard deviation with ±?

**Limitations:**

In my opinion, the authors have not entirely addressed the limitations of their work, specifically, in relation to my comment in the "Weaknesses" section


I set my assessment to "borderline reject" but my opinion can change based on the author's answers and comments

---

> ### Author Rebuttal · Authors · 2023-08-09
>
> Thank you for your insightful comments and suggestions. Please read our response below and let us know if you have more questions.
>
> **Q1**: NERE DSM relies on co-crystal structures for training and co-crystal/docked structures for binding prediction of a new protein-ligand / Ab-Ag pair, which can be limiting in terms of experimental/computational resources.
>
> In terms of computational resources, modern molecular docking tools (based on neural networks) are much faster than traditional docking tools like AutoDock Vina and ZDOCK. For example, PLANET [1] is a neural network model for protein-ligand docking. On our CASF test set, it takes only about 0.1 second for PLANET to dock one ligand (on average). GeoDock [2] is another neural network model for protein-protein docking. On our SAbDab test set, it takes about 0.1 second to dock one antibody to its antigen. Using these neural network based docking models, we can screen over millions of compounds/antibodies per day. We also evaluated the performance of our model when using the structures docked by PLANET and GeoDock. As shown in the table below, there is only a small decrease in Pearson correlation but the model remains accurate. Therefore, we argue that computational resources are not limiting factors to our unsupervised binding energy prediction framework.
>
> |        | Slow Docking (Vina/ZDOCK) | Ultrafast Docking (PLANET/GeoDock) |
> |----|---|----|
> | CASF | 0.656 ± 0.012 | 0.616 ± 0.009 |
> | SAbDab | 0.360 ± 0.051 | 0.350 ± 0.069  |
>
> In terms of experimental resources, our method requires co-crystal structures for training. We agree that it can be a limiting factor but we can alleviate this issue by utilizing a larger set of protein-protein complexes to pre-train our model. There are 208,347 co-crystal structures in the Protein Data Bank in total and its growth is about 10,000 per year. We will explore this direction in our future work.
>
> **Q2**: Docking speed of AutoDock Vina, ZDOCK, and other docking models.
>
> The docking speed of AutoDock Vina and ZDOCK is 25 second and 20 second per complex (using a 64-CPU server). The docking speed is much faster for neural network docking tools like GeoDock and PLANET, with approximately 0.1 second per complex.
>
> **Q3**: Are there no cases where affinity data is available together with co-crystal data so both supervised and unsupervised models could be trained on the same dataset?
>
> For antibodies, there are only 566 co-crystal structures that have affinity labels. They are all included in the test set and could not be used to train our model. For small molecules, the co-crystal structures in our training set actually come with affinity labels, though we did not use the affinity labels to train our model. For a fair comparison, we trained a supervised model on the same training set and model architecture as NERE DSM, but using affinity labels and a regression training objective. This supervised model achieves an average Pearson correlation of 0.734 ± 0.012 on the CASF test set, which is lower than TankBind or IGN because it uses less training data. This supervised model is better than NERE DSM but the margin is relatively small (0.734 vs 0.656). In this case, our unsupervised method is able to recover nearly 90% of the supervised model performance.
>
> **Q4**: Why were only the top 50 antigen residues chosen as epitopes instead of all residues within a certain distance, as done for the protein in the small molecule case?
>
> We found that the latter strategy yielded a lower performance (Pearson R = 0.317) because the number of epitope residues are quite uneven when using a distance threshold. The performance becomes better when using a fixed number of epitope residues.
>
> **Q5**: Details on the nature of the test splits
>
> The current train/test split is based on IID, but we filtered the training set so that the same ligand/antibody/antigen does not appear in both training and test sets. To further study the impact of train/test split strategy, we also train our model with sequence similarity split. In this case, we remove all training instances whose proteins/antibodies/antigens have 40% similarity to the test set. As shown in the following table, we find that the average performance of NERE is not impacted by similarity split (0.508 vs 0.503), which confirms that our model is not making predictions simply based on similarity.
>
> |        | Original split | Similarity split |
> |----|----|-----|
> | CASF (Crystal) | 0.656 ± 0.012 | 0.634 ± 0.009 |
> | SAbDab (Crystal) | 0.360 ± 0.051 | 0.372 ± 0.038 |
> | Average | 0.508 | 0.503 |
>
> **Q6**: Could you report a global metric, for example, the average difference in the test set between the energy of CDR3 residues versus non-CDR3 residues (plus statistical significance)?
>
> The average difference between the energy of a CDR3 residue and a non-CDR3 residue is -2.18, with a p-value of 5.8e-60 under student t-test. From the rebuttal Figure 1g, we can clearly see that the average difference is shifted towards negative range. Therefore, we conclude that the model pays more attention to CDR3 residues than non-CDR3 residues.
>
> **Q7**: Experiments were repeated with 5 random seeds but only the average values were reported in table 1, could you also report the standard deviation with ±?
>
> The standard deviation is already reported in table 1 as subscripts (e.g., $0.656_{.011}$ means 0.656 ± 0.011).
>
> **Q8**: In the antibody-antigen case, are the supervised model pre-trained?
>
> Yes, the supervised $\mathrm{FANN_{transfer}}$ model is pre-trained on the SKEMPI protein-protein binding database, which contains approximately 6000 binding affinity data points. This model also uses the pre-trained ESM-2 protein language model embedding for residue features.
>
> Reference
> 1. Zhang et al. Planet: A multi-objective graph neural network model for protein-ligand binding affinity prediction. bioRxiv 2023
> 2. Chu et al. Flexible Protein-Protein Docking with a Multi-Track Iterative Transformer. bioRxiv 2023

---

> > ### Comment · Reviewer_3x52 · 2023-08-14
> > **Thanks**
> >
> > I thank the authors for their thoughtful responses. The authors addressed all my questions.
> > I have 2 remaining comments:
> > 1. Regarding response to Q1: "In terms of experimental resources, our method requires co-crystal structures for training. We agree that it can be a limiting factor but we can alleviate this issue by utilizing a larger set of protein-protein complexes to pre-train our model. There are 208,347 co-crystal structures in the Protein Data Bank in total and its growth is about 10,000 per year. We will explore this direction in our future work." There is no guarantee that pretraining on PPI data would help, particularly for the Ab-Ag prediction case. In any case, pretraining does not remove the model's limitation of only being trainable with co-crystal structures. Any thoughts on adapting the model to be trainable with docked structures?
> > 2. Regarding Q5, was there a reasoning/justification behind the 40% similarity cutoff?

---

> > > ### Author Response · Authors · 2023-08-15
> > > **Thanks**
> > >
> > > Thank you for your thoughtful comments.
> > >
> > > Q1: Any thoughts on adapting the model to be trainable with docked structures?
> > >
> > > Our model can be directly trained on docked structures if they are accurate (close to crystal structures). In the case of protein-ligand binding, we can leverage docked structures from the CrossDocked2020 dataset [1], which has 22.5 million docked poses. We can use their binding pose classification model (accuracy = 0.95 AUC) to evaluate the accuracy of each docked structure and add only accurate structures to our training set. The same approach works for antibody-antigen binding. For example, we can use state-of-the-art protein complex structure prediction models like AlphaFold-multimer (AFM) [2] to generate a large set of antibody-antigen complexes. AFM gives a confidence score for each predicted structure that indicates how likely a docked pose is correct. The docking success rate of AFM on the standard antibody-antigen docking benchmark [3] is 28.4%, but the success rate becomes 80% if we only look at docked complexes with confidence above 0.8. Therefore, we can add these highly confident complexes into the training set without contaminating it with too many incorrect poses.
> > >
> > > Q2: Was there a reasoning/justification behind the 40% similarity cutoff?
> > >
> > > We used the 40% similarity cutoff because we saw that some prior work on antibody generation [4, 5] used the same similarity cutoff for their experiments. In addition, we found that most of the similar instances are filtered at the 70-90% threshold. For example, in the protein-ligand case, the original training set had 4806 complexes. With the 90% cutoff, the training set size became 3352. With the 40% cutoff, the training set size became 3064. The filtering speed was 1450 (per 10%) in the beginning and 60 (per 10%) after that. Based on these two observations, we thought that the 40% similarity cutoff was reasonable.
> > >
> > > References
> > > 1. Francoeur et al. 3D Convolutional Neural Networks and a CrossDocked Dataset for Structure-Based Drug Design, 2020
> > > 2. Evans et al. "Protein complex prediction with AlphaFold-Multimer." biorxiv 2021
> > > 3. Guest et al. "An expanded benchmark for antibody-antigen docking and affinity prediction reveals insights into antibody recognition determinants." Structure 2021
> > > 4. Jin et al. "Iterative refinement graph neural network for antibody sequence-structure co-design." ICLR 2022
> > > 5. Luo et al. Antigen-Specific Antibody Design and Optimization with Diffusion-Based Generative Models for Protein Structures. NeurIPS 2022

---

> > > > ### Author Response · Authors · 2023-08-19
> > > > **Additional experiment with docked structures**
> > > >
> > > > Thank you again for your insightful comments. We understand your concern about the limitation of using co-crystal structures for training. Based on your suggestion, we tried training our model on docked structures in the context of protein-ligand binding because there is a large number of docked structures readily available in the CrossDock 2020 dataset [1]. Specifically, we selected 45000 docked complexes that are close to crystal structures (with RMSD < 2Å, as suggested by CrossDock). We further filter this dataset by removing any proteins with over 40% sequence similarity to any of the test set proteins. After this filtering step, our final training set has approximately 30000 docked complexes spanning 2800 different proteins. We compare our model trained on this CrossDock data against our original model trained on the PDBBind co-crystal complexes. As shown in the following table, training on a larger set of high-quality docked structures slightly improves the model performance. Though the improvement is not that significant, it still demonstrates that our model is not restricted to only co-crystal structures. We hope this experiment addresses your concerns. Please let us know if you have any additional questions before the discussion period ends. Thank you very much!
> > > >
> > > > |    | CASF (Similarity split) |
> > > > |:-------|:---------|
> > > > | NERE trained on PDBBind (crystal) |      0.634 ± 0.009      |
> > > > | NERE trained on CrossDock         |      **0.640 ± 0.006**      |

---

> > > > > ### Comment · Reviewer_3x52 · 2023-08-21
> > > > > **Thanks for your responses**
> > > > >
> > > > > Thank you for your thoughtful comments and additional experiments! Your response has made me reconsider my review and I will be raising my score.

---

### Official Review · Reviewer_j9Y6 · 2023-07-06

**Soundness:** 3 good
**Presentation:** 3 good
**Contribution:** 3 good
**Rating:** 6
**Confidence:** 4

**Summary:**

The paper introduces an unsupervised learning approach for predicting protein-ligand binding energy, called NERE (Neural Equivariant Rotation Estimation). The authors employ SE(3) denoising score matching to train an energy-based model on a dataset of unlabeled protein-ligand complexes. The model's effectiveness is demonstrated through evaluations on protein-ligand and antibody-antigen binding affinity benchmarks from PDBBind and the Structural Antibody Database (SAbDab).

**Strengths:**

1. The paper innovatively applies SE(3) score matching to train an energy-based model, positioning it as an unsupervised energy ranking model.
2. The experimental results highlight the model's effectiveness, particularly when compared with other unsupervised models or physics-based models.
3. The paper is well-structured and easy to comprehend, making the proposed method and its implications clear to the reader.

**Weaknesses:**

1. The proposed method only considers rigid transformations (translations & rotations) in the diffusion process, neglecting the degrees of freedom in small molecule ligands, such as additional rotatable bonds.
2. The method employs a residue-level representation for proteins, which might omit fine-grained information that could be crucial for accurate binding prediction.

**Questions:**

1. Why was the SE(3) diffusion process chosen over the standard elucidating diffusion process? While the standard diffusion process may generate nonsensical conformations, the proposed diffusion process might not explore the entire conformation space.
2. In section 4.1, is the center of mass, denoted as $\mu$, computed based on atomic weight?
3. How does the model perform in the "Docked structure" scenario when the docking error is high?
4. Why does contrastive learning underperform compared to using score matching to train the energy-based model? Further elaboration on this would be beneficial.
5. Are there any experiments conducted on the docking benchmark of the CASF test set? It seems that the proposed method is more suitable for identifying the correct binding pose (ranking among multiple poses of a specific molecule) rather than ranking between binding poses of multiple molecules.
6. Different noise levels appear to lead to different energy functions, while only the minimal noise level can approximate the energy function of the real data distribution. How does the proposed method handle multiple noise scales?

**Limitations:**

The proposed method appears to be sensitive to docking error. However, in real-world scenarios, finding the true binding pose is a challenging problem. This sensitivity could limit the method's practical applicability.

---

> ### Author Rebuttal · Authors · 2023-08-09
>
> Thank you for your insightful comments and suggestions. Please read our response below and let us know if you have more questions.
>
> **Q1**: The proposed method only considers rigid transformations (translations & rotations) in the diffusion process, neglecting the degrees of freedom in small molecule ligands, such as rotatable bonds. The method employs a residue-level representation for proteins, which might omit fine-grained information that could be crucial for accurate binding prediction.
>
> We agree that it is beneficial to model the flexibility of ligands and incorporate the side-chain information of proteins. Given the limited time, we partially address this issue by including all protein atoms (including side-chains) as input instead of only $C_\alpha$ atoms. This modification only changes the input of our model while the model architecture and NERE DSM objective remains the same. Our results suggest that including side-chain atoms substantially improves the performance of our method on some of the datasets. Please read our global response above and the rebuttal Figure 1e for details. In the future, we plan to model the flexibility of rotatable bonds and protein side-chain atoms by introducing a side-chain DSM loss, which adds rotation noise on the rotatable bonds and side-chain atoms and use NERE to predict the rotation noise.
>
> **Q2**: Why was the SE(3) diffusion process chosen over the standard Euclidean diffusion process? While the standard diffusion process may generate nonsensical conformations, the proposed diffusion process might not explore the entire conformation space.
>
> We have already included results of the standard Euclidean diffusion process in our original manuscript (named as Gauss DSM in Table 1). As shown in the table below, our results suggest that SE(3) diffusion is better than Euclidean diffusion on both test sets.
>
> |   | CASF (crystal) | CASF (docked) | SAbDab (crystal) | SAbDab (docked) |
> |----------------------|----------------|---------------|------------------|-----------------|
> | Eucliean diffusion | 0.638 ± 0.017  | 0.632 ± 0.016 | 0.335 ± 0.038  | 0.330 ± 0.022   |
> | SE(3) diffusion | 0.656 ± 0.012  | 0.651 ± 0.011 | 0.361 ± 0.051 | 0.360 ± 0.051   |
>
> **Q3**: In section 4.1, is the center of mass computed based on atomic weight?
>
> For small molecule ligands, the center of mass is set as a constant for all atom types because the atomic weight of the most common atoms are quite close (e.g., C=12, N=14, O=16). For antibodies, the center of mass is computed based on amino acid weight because they are more diverse (e.g., Glycine = 57, tryptophan = 204). To validate this modeling choice, we run our model with either constant weight or true atomic weight. As shown in the table below, the effect of using atomic weight is negligible for small molecules but quite helpful for antibodies.
>
> |  | CASF (crystal) | SAbDab (crystal) |
> |---------------|----------------|------------------|
> | Constant weight  | 0.656  | 0.340  |
> | True atomic/residue weight | 0.657  | 0.352 |
>
> **Q4**: How does the model perform in the "Docked structure" scenario when the docking error is high?
>
> Our model performance is quite robust to docking error. In the antibody case, the RMSD of ZDOCK is around 19.4 but the performance of NERE DSM under docked structures is quite close to crystal structures (Pearson correlation: 0.361 vs 0.360; Spearman correlation: 0.385 vs 0.363).
>
> **Q5**: Why does contrastive learning underperform denoising score matching (DSM)? Further elaboration on this would be beneficial.
>
> Contrastive learning is a simpler objective than DSM because it only teaches the model to assign lower energy to crystal structures than perturbed structures. In contrast, the DSM objective not only teaches the model to assign lower energy to crystal structure, but also to reconstruct the original ligand/antibody pose from perturbed structures. It requires the model to leverage more geometric information in order to succeed in this more difficult task.
>
> **Q6**: Are there any experiments conducted on the docking benchmark of the CASF test set? It seems that the proposed method is more suitable for identifying the correct binding pose (ranking among multiple poses of a specific molecule) rather than ranking between binding poses of multiple molecules.
>
> We agree that the docking experiment is also relevant to our approach. Our current focus is protein-ligand binding affinity prediction but we will include this experiment in our future work. Thank you for your suggestions.
>
> **Q7**: Different noise levels appear to lead to different energy functions, while only the minimal noise level can approximate the energy function of the real data distribution. How does the proposed method handle multiple noise scales?
>
> In our implementation, we use multiple noise scales to train our model. In each training step, we first randomly sample a noise scale $\sigma \sim [0, 10]$ for the rotation noise and then sample a rotation vector $\omega \sim \mathcal{N}_\mathrm{SO(3)}(\sigma)$. This dynamic sampling strategy allows us to sample noise of both small and large scales. The small-scale noise helps the model to approximate the real data distribution, while the large-scale noise allows the model to be trained on a diverse range of perturbed structures.

---

> > ### Comment · Reviewer_j9Y6 · 2023-08-16
> >
> > Thanks for your response. I appreciate your efforts to address my concerns, but I remain concerned aboout the reasonability of varying noise levels. Specifically, introducing different noise levels into the data distribution results in diverse corrupted distributions, each of which may correspond to a distinct energy function. An extreme case would be the addition of extremely high levels of noise, which could obliterate all information from the original distribution. Yet, in the training objective, a single energy function is employed to model different noise levels simultaneously. Could this approach have some potential problems?

---

> > > ### Author Response · Authors · 2023-08-16
> > > **Multiple noise level**
> > >
> > > Thank you for your comment. Our approach is inspired by Noise Conditional Score Network (NCSN, Yang et al 2020), where they sampled different noise levels during training. The difference between our approach and NCSN is that our model is not conditioned on the noise level. As a result, our current training objective is matching our energy function to a mixture distribution, i.e., a mixture of perturbed distributions given by different noise levels. This mixture distribution is a valid distribution (analogous to Gaussian mixture models), but could be sub-optimal because it is not close enough to the original data distribution. Currently, we are not adding extreme levels of noise and that's why our model performs reasonably well.
> > >
> > > In principle, there are two solutions to address your concern. First, we can train our model with only one (small) noise level. We can tune this hyper-parameter on the validation set to find the optimal noise level. Second, we can condition the energy function on the noise level by appending the input with some encoding of $\sigma$. Similar to NCSN, we are learning multiple energy functions (with shared parameters) at the same time, each corresponds to one noise level. At test time, we can choose the one with the best cross-validation performance. We will explore this direction in our future work.

---

> > > > ### Comment · Reviewer_j9Y6 · 2023-08-17
> > > >
> > > > Thanks for your response. My concern is well addressed.

---

### Official Review · Reviewer_X8hw · 2023-07-07

**Soundness:** 3 good
**Presentation:** 3 good
**Contribution:** 3 good
**Rating:** 6
**Confidence:** 3

**Summary:**

This paper introduces NERE, an unsupervised method for predicting protein-ligand binding affinity. The authors propose a generative modeling approach that utilizes an energy-based model (EBM) to capture the characteristics of protein-ligand complexes. The EBM is trained by maximizing the log-likelihood of crystal structures from the PDB database. NERE incorporates random rotations and translations of the separate parts of the molecule complexes and predicts the optimal rotation and translation as the EBM's output. To train the EBM, SE(3) denoising score matching is employed. The effectiveness of the proposed approach is evaluated through experiments on both protein-ligand and protein-protein binding. The results reveal a positive correlation between the predicted energy and binding affinity. The authors demonstrate that their method surpasses all unsupervised baselines and even outperforms supervised baselines in the case of antibodies.

**Strengths:**

Strengths of the review paper:

- The paper introduces a novel approach for unsupervised protein-ligand binding energy prediction by utilizing an equivariant rotation prediction network. This innovative approach contributes to the existing body of knowledge in the field.

- The paper proposes a unique method to predict rotation and translation as part of an unsupervised data augmentation technique. This approach enhances the understanding and modeling of protein-ligand interactions, as well as the correlation with molecule interaction energies.

- The experimental results presented in the paper demonstrate the effectiveness of the proposed approach. The approach outperforms all unsupervised baselines, showcasing its superiority in predicting protein-ligand binding energies. Notably, the method even surpasses supervised baselines in the case of antibodies, indicating its potential for practical applications.

- The paper exhibits clear and well-written content. The authors effectively convey technical concepts and methods, ensuring the readers can grasp the details of the proposed approach and its implementation. The organization of the paper enhances readability and comprehension.

- The authors demonstrate their awareness of the limitations of their work.

**Weaknesses:**

- The paper lacks visualizations of the correlation between the predicted energies and binding affinities. Visual representations, such as scatter plots or correlation matrices, would enhance the understanding of the relationship and provide a clearer picture of the results. Additionally, providing information about the distribution of outliers would give insights into the robustness of the proposed approach.
- The authors could consider spearman correlation and p-value as additional evaluation metrics, as the ranking between different ligands might be more useful for drug discovery and virtual screening instead of absolute binding affinity values.
- I am curious about the potential impact of the proposed unsupervised method on downstream tasks when serving as a pre-training strategy. It would be valuable to investigate whether the learned representations from the unsupervised approach can be beneficial for supervised downstream tasks.
- The paper lacks an explicit discussion on the significance and relevance of an unsupervised method in the context of protein-ligand binding energy prediction. It would be beneficial to design experiments or provide theoretical arguments that illustrate how the utilization of unlabeled data, in comparison to traditional supervised methods, contributes to the understanding and performance of the model. Demonstrating the utility of unsupervised learning in this specific area would provide a stronger rationale for the proposed approach.


**Questions:**

Please see weaknesses.


**Limitations:**

None.

---

> ### Author Rebuttal · Authors · 2023-08-09
>
> Thank you for your insightful comments and suggestions. Please read our response below and let us know if you have more questions.
>
> **Q1**: The paper lacks visualizations of the correlation between the predicted energies and binding affinities.
>
> In rebuttal Figure 1a, we visualize the correlation between the predicted energies and binding affinities on the small molecule test set (CASF) and the antibody test set (SAbDab). We did not find significant outliers in either crystal or docked settings.
>
> **Q2**: The authors could consider spearman correlation and p-value as additional evaluation metrics
>
> In the rebuttal Table 1, we report the spearman correlation and p-value of all methods on both test sets. Our model (NERE DSM) remains the best under both metrics.
>
> **Q3**: I am curious about the potential impact of the proposed unsupervised method on downstream tasks when serving as a pre-training strategy. It would be valuable to investigate whether the learned representations from the unsupervised approach can be beneficial for supervised downstream tasks.
>
> We adopt an HER2 antibody affinity maturation dataset [1] as an additional benchmark to investigate the utility of NERE DSM as a pre-training strategy. This dataset has 422 antibody sequences with experimental binding affinities against HER2. The pre-trained NERE DSM model achieved an average spearman correlation of 0.230 on this dataset, while the best supervised baseline ($\mathrm{FANN_{transfer}}$) only yields 0.090. To further improve the performance, we selected a subset of the HER2 data to fine-tune the pre-trained NERE DSM model and the supervised $\mathrm{FANN_{transfer}}$ model. Specifically, we split the HER2 dataset into a training set of $N$ data points and use the rest of the dataset for testing. As shown in rebuttal Figure 1f, NERE DSM consistently outperforms $\mathrm{FANN_{transfer}}$ when fine-tuned by $N=50, 100, 150, 200$ data points. These results suggest that NERE DSM is a more effective pre-training strategy than supervised pre-training.
>
> **Q4**: The paper lacks an explicit discussion on the significance and relevance of an unsupervised method in the context of protein-ligand binding energy prediction. It would be beneficial to design experiments that illustrate how the utilization of unlabeled data, in comparison to traditional supervised methods, contributes to the performance of the model.
>
> Indeed, the performance of unsupervised methods is much lower than supervised models on the CASF test set (PDBBind core set). However, the supervised models are trained and evaluated on the same PDBBind database and it is still unclear whether they generalize well to new molecules with different structures. To this end, we adopt the free energy perturbation (FEP) benchmark [2] (developed by Merck) as an additional independent test set. It has eight protein targets (cdk8, cmet, eg5, hif2a, pfkfb3, shp2, syk, and tnks2) and 264 ligands in total. As shown in Figure 1b, molecules in this test set have a different distribution from the PDBBind training set. Therefore, a model needs to generalize to a different chemical space to perform well on this test set.
>
> We applied NERE DSM to this FEP test set, with a small modification in the input layer to include protein side-chain atoms (see our global response above). We calculate the Spearman correlation ($R_S$) between its predicted binding energy and experimental binding affinity for each of the eight targets and report the average correlation as our evaluation metric. Importantly, all hyper-parameters are selected using only the PDBBind validation set to ensure this experiment is a blind, unbiased evaluation. As shown in Figure 1c, our model outperforms not only the unsupervised models (contrastive learning and Gaussian DSM) but also the supervised models. The performance is much better than PLANET & IGN (0.380 vs 0.314 and 0.252) and slightly better than TankBind (0.380 vs 0.376). The Spearman correlation for each target is shown in Figure 1d. Overall, these results suggest that unsupervised models are more robust than supervised models when there is distributional shift (a common problem when models are deployed in the wild). Therefore, unsupervised models can be also useful for protein-ligand binding, even though there is a fair amount of labeled affinity data.
>
> References
> 1. Shanehsazzadeh et al., Unlocking de novo antibody design with generative artificial intelligence, biorxiv 2023
> 2. Schindler et al., Large-scale assessment of binding free energy calculations in active drug discovery projects, 2020

---

> > ### Comment · Reviewer_X8hw · 2023-08-17
> > **Thank you for your response**
> >
> > Thank you for your detailed response. I think the results are more convincing than before and this an interesting task so I have raised my score accordingly.

---

> > > ### Author Response · Authors · 2023-08-17
> > > **Thank you**
> > >
> > > Thank you very much. Your comments are very beneficial to us and we sincerely appreciate your effort in reviewing our paper.

---

### Author Rebuttal · Authors · 2023-08-09

We want to thank all reviewers for their valuable comments and suggestions. We would like to summarize three main results here in response to some common questions/suggestions. The results are included in the attached PDF file (rebuttal Table 1 and Figure 1a-h).

### Additional metrics

As suggested by Reviewer X8hw and UbeA, we report Spearman correlation and p-values as additional metrics in rebuttal Table 1. We could not report root mean square error (RMSE) or mean absolute error (MAE) because our model does not predict absolute affinity values. Shifting or scaling $E(A, X)$ by any constant will be equally optimal under the DSM objective.

### Additional benchmark datasets

We include two additional benchmark datasets to incorporate reviewer X8hw’s suggestions. First, we adopt an HER2 antibody affinity maturation dataset [1] to investigate the utility of NERE DSM as a pre-training strategy. This dataset has 422 antibody sequences with experimental binding affinities against HER2. We report the Spearman correlation between the predicted binding energy and experimental binding affinity on this dataset. To investigate whether the learned representations from the unsupervised approach can be beneficial for supervised downstream tasks, we randomly sample a small number of HER2 data to fine-tune the pre-trained NERE model. Results on this dataset are shown in rebuttal Figure 1f. When fine-tuned on the same number of HER2 data, we find that representations learned by NERE outperform representations learned under supervised pre-training ($\mathrm{FANN_{transfer}}$).

Second, we adopt a free energy perturbation (FEP) benchmark [2] (developed by Merck) to investigate the utility of unsupervised learning in the context of protein-ligand binding. It has eight protein targets (cdk8, cmet, eg5, hif2a, pfkfb3, shp2, syk, and tnks2) and 264 ligands in total. We calculate the Spearman correlation ($R_S$) between the predicted binding energy and experimental binding affinity for each of the eight targets and report the average correlation for each method. Results on this dataset are shown in rebuttal Figure 1b-d. Our main finding is that NERE outperforms supervised baselines on this FEP test set. These results suggest that unsupervised models can be also useful for protein-ligand binding even though there is a fair amount of labeled affinity data. The experimental setup for these two experiments are detailed in our response to reviewer X8hw.

### Including side-chain atoms into the model

Several reviewers have raised concerns that the current model does not model the flexibility of proteins/ligands and the side chains of proteins. Given the limited time, we partially address these concerns by including all protein atoms (backbone + side chains) as input instead of only $C_\alpha$ atoms. This modification only changes the input of our model while the model architecture and NERE DSM objective remains the same. We refer to this extended model as NERE DSM (all-atom) and the original model as NERE DSM ($C_\alpha$-only).

As shown in rebuttal Figure 1e, we found that adding side-chain information gave a similar performance on the original CASF and SAbDab test sets, but substantially improved model performance on the new FEP and HER2 datasets. On the FEP test set, the all-atom model achieved $R_S=0.380$ while the $C_\alpha$-only model had $R_S=0.192$. On the HER2 test set, the all-atom model achieved $R_S=0.230$ while the $C_\alpha$-only model had only $R_S=0.043$. Overall, these results suggest that incorporating side-chain information can be beneficial for protein-ligand/antibody-antigen binding energy prediction. In the future, we plan to model the flexibility of rotatable bonds and protein side-chain atoms by introducing a side-chain DSM loss, which adds rotation noise on the rotatable bonds and side-chain atoms and use NERE to predict the rotation noise.

### References
1. Shanehsazzadeh et al., Unlocking de novo antibody design with generative artificial intelligence, biorxiv 2023
2. Schindler et al., Large-scale assessment of binding free energy calculations in active drug discovery projects, 2020

---

### Decision · Program_Chairs · 2023-09-21

**Decision:**

Accept (poster)

**Comment:**

The paper proposes an unsupervised method for binding classification using denoising score matching. I like this paper. The problem considered is a pertinent one within drug and protein design. The paper is well written, the results are strong and the approach is novel. The additional numerical results presented during the rebuttal period further corroborate the claims of good performance.